# Juvenile Arthritis Patients Suffering from Chronic Inflammation Have Increased Activity of Both IDO and GTP-CH1 Pathways But Decreased BH4 Efficacy: Implications for Well-Being, Including Fatigue, Cognitive Impairment, Anxiety, and Depression

**DOI:** 10.3390/ph12010009

**Published:** 2019-01-08

**Authors:** Gerdien A. H. Korte-Bouws, Eline Albers, Marije Voskamp, Hendrikus Hendriksen, Lidewij R. de Leeuw, Onur Güntürkün, Sytze de Roock, Sebastiaan J. Vastert, S. Mechiel Korte

**Affiliations:** 1Division of Pharmacology, Utrecht Institute for Pharmaceutical Sciences (UIPS), Utrecht University, Faculty of Science, Universities 99, 3584 CG Utrecht, The Netherlands; G.A.H.Korte@uu.nl (G.A.H.K.-B.); e.g.albers@students.uu.nl (E.A.); m.j.voskamp@students.uu.nl (M.V.); H.Hendriksen@uu.nl (H.H.); l.r.deleeuw@uu.nl (L.R.d.L.); 2Department of Biopsychology, Faculty of Psychology, Ruhr-Universität Bochum, Universitätsstraße 150, D-44780 Bochum, Germany; onur.guentuerkuen@ruhr-uni-bochum.de; 3Paediatric Rheumatology, Wilhelmina Children’s Hospital, University Medical Center Utrecht, Lundlaan 6, 3584 EA Utrecht, The Netherlands; S.deRoock@umcutrecht.nl (S.d.R.); B.Vastert@umcutrecht.nl (S.J.V.)

**Keywords:** arthritis, inflammation, sickness behavior, fatigue, biomarkers, indoleamine-2,3-dioxygenase (IDO), guanosinetriphosphate–cyclohydrolase-1 (GTP–CH1), tetrahydrobiopterin (BH4), tryptophan, kynurenine, tyrosine, neopterin, phenylalanine

## Abstract

Juvenile idiopathic arthritis (JIA) represents joint inflammation with an unknown cause that starts before the age of 16, resulting in stiff and painful joints. In addition, JIA patients often report symptoms of sickness behavior. Recent animal studies suggest that proinflammatory cytokines produce sickness behavior by increasing the activity of indoleamine-2,3-dioxygenase (IDO) and guanosinetriphosphate–cyclohydrolase-1 (GTP–CH1). Here, it is hypothesized that inflammation in JIA patients enhances the enzymatic activity of IDO and GTP-CH1 and decreases the co-factor tetrahydrobiopterin (BH4). These compounds play a crucial role in the synthesis and metabolism of neurotransmitters. The aim of our study was to reveal whether inflammation affects both the GTP-CH1 and IDO pathway in JIA patients. Serum samples were collected from twenty-four JIA patients. In these samples, the concentrations of tryptophan (TRP), kynurenine (KYN), tyrosine (TYR), neopterin, and phenylalanine (PHE) were measured. An HPLC method with electrochemical detection was developed to quantify tryptophan, kynurenine, and tyrosine. Neopterin and phenylalanine were quantified by ELISA. The KYN/TRP ratio was measured as an index of IDO activity, while the PHE/TYR ratio was measured as an index of BH4 activity. Neopterin concentrations were used as an indirect measure of GTP-CH1 activity. JIA patients with high disease activity showed higher levels of both neopterin and kynurenine, and a higher ratio of both KYN/TRP and PHE/TYR and lower tryptophan levels than clinically inactive patients. Altogether, these data support our hypothesis that inflammation increases the enzymatic activity of both IDO and GTP-CH1 but decreases the efficacy of the co-factor BH4. In the future, animal studies are needed to investigate whether inflammation-induced changes in these enzymatic pathways and co-factor BH4 lower the levels of the brain neurotransmitters glutamate, noradrenaline, dopamine, serotonin, and melatonin, and consequently, whether they may affect fatigue, cognition, anxiety, and depression. Understanding of these complex neuroimmune interactions provides new possibilities for Pharma-Food interventions to improve the quality of life of patients suffering from chronic inflammation.

## 1. Introduction

Juvenile idiopathic arthritis (JIA) is a chronic inflammatory disease that has an onset before the age of 16 and is characterized by periods of disease flares [1]. The core symptoms of JIA are pain, swelling, stiffness, fever, rash, and limping [2]. JIA can affect one joint or many, and while some patients experience these symptoms for only a short period of time, others can suffer from JIA their whole lives [2].

Remarkably, a quarter of the JIA patients suffer from severe fatigue, and they prioritize fatigue within their main problems [3,4]. Fatigue, which is a core symptom of sickness behavior, is reported by patients as extreme, overwhelming, unremitting, and unrelated to physical activity [4,5,6]. Unfortunately, fatigue is difficult to evaluate and quantify because of its subjective nature and varied appearances (e.g., cognitive, sleep/rest, physical, emotional) [4,5,6,7]. In inflammatory disorders, central fatigue is often observed that comprises several dimensions, namely physical fatigue (i.e., reduced performance in physical tasks), mental fatigue (i.e., reduced performance in cognitive tasks), and the lack of motivation (i.e., reduced effort to obtain hedonic reward) [8]. Therefore, it is not surprising that JIA patients suffering from central fatigue report it as under-recognized and rarely treated [5]. 

Next to fatigue, more brain-related symptoms may develop. In the long term, arthritis patients often have inflammation-induced cognitive impairments [9]. Furthermore, in children with JIA, a significantly higher rate of anxiety and depression can be observed compared to a healthy control group [10].

Remarkably, a multi-modal MRI study in rheumatoid arthritis (RA) patients showed that high levels of inflammation were associated with more connections between the inferior parietal lobule (IPL), medial prefrontal cortex, and multiple brain networks (including the dorsal attention network, salience network, and medial visual network), as well as reduced IPL grey matter. These patterns of connectivity predicted fatigue, pain, and cognitive dysfunction [11].

Most recently, a review in the journal *The Lancet* discussing the link between inflammation and the comorbidity of rheumatoid arthritis and depression [12] showed that psychoneuroimmunology research is finally being taken seriously, and that it has arrived in the domain of the medical world.

Coming from studying disparate disorders, i.e., immune diseases and brain disorders, elements that were previously considered to be domains of one discipline are now discovered in the other. There is a rapidly growing amount of evidence demonstrating a strong bidirectional signaling between the immune system and the brain. For example, a proinflammatory response, characterized by high blood levels of the following cytokines: TNF-α; IL-1; and IL-6 [13], affects monoamine neurotransmitter systems to induce sickness behavior [14,15], including the core symptom of fatigue [4]. In agreement, proinflammatory conditions during rheumatoid arthritis (RA), multiple sclerosis (MS), and even cancer go hand in hand with fatigue, cognitive impairment, and depressive symptoms [4,16,17]. In agreement, it has been reported that depression and fatigue are often associated with increased levels of proinflammatory cytokines [18,19].

Previously, a strong relation between fatigue and pain was shown in JIA patients, which was the strongest for females [3]. The relationship between inflammatory activity and disease activity was less clear [20,21]. Although some biologicals have beneficial effects on fatigue, often a large part of the fatigue remains [4,21]. Sometimes, paradoxical drug effects can be observed. For example, it has been reported that the immunosuppressant methotrexate (MTX) produced less fatigue in patients with rheumatoid arthritis [21], but more fatigue was observed after MTX treatment in another study [22].

These unclear and inconsistent data leave a perfect opportunity for us to gain more knowledge on the relationship between inflammation, disease activity, and possible alterations in enzymatic activity of indoleamine-2,3-dioxygenase (IDO) and guanosinetriphosphate–cyclohydrolase-1 (GTP–CH1), which are involved in the synthesis of neurotransmitters. Previously, it was shown that a chronic low-grade inflammation in elderly people was associated with altered tryptophan and tyrosine metabolism, which was suggested to be involved in disturbed mental well-being [23]. Remarkably, the same study showed that inflammation was correlated with higher levels of both neopterin and kynurenine, lower levels of tryptophan, and a higher KYN/TRP ratio [23]. In the present study, we hypothesize that increased chronic inflammation affects the biosynthesis and metabolism of neurotransmitters, which may have consequences for well-being, including fatigue, cognition, anxiety, and depression.

Therefore, in the present study, several precursors and amino acids are measured as a reflection of enzyme activity and co-factor BH4 efficacy. To this end, concentrations of neopterin, kynurenine, tryptophan, tyrosine, and phenylalanine were measured in the serum of JIA patients who have high disease activity or are clinically inactive and receive MTX or no drug treatment. 

## 2. Results

### 2.1. Disease Activity and MTX Treatment

The disease activity of the participants was classified according to the clinical juvenile arthritis disease activity score (cJADAS). Clinically inactive (IA) patients were defined as cJADAS <0.5, whereas high disease activity (HDA) was set at cJADAS ≥10.0. (see Table 1). The corresponding C-reactive protein (CRP) values were in line with the disease activity scores. Remarkably, MTX treatment significantly (*p* < 0.01) lowered CRP levels during high disease activity, while the cJADAS score was not significantly reduced but was still significantly higher (*p* < 0.05) than that of clinically inactive patients. There were neither significant effects of being male or female nor of MTX-treatment on tryptophan (TRP), kynurenine (KYN), neopterin, phenylalanine (PHE), tyrosine (TYR), and the PHE/TYR ratio. Therefore, these data were pooled into two groups, more precisely, HDA (*n* = 12) and IA (*n* = 12), except for the KYN/TRP ratio during high disease activity (HDA), because MTX treatment lowered it significantly (*p* < 0.05).

### 2.2. GTP–CH1 Activity

#### 2.2.1. Neopterin

Neopterin concentrations in the high disease activity (HDA) JIA group were significantly higher (*p* < 0.01) compared to levels in clinically inactive (IA) patients (see Figure 1).

#### 2.2.2. Phenylalanine and Tyrosine

Disease activity affected neither phenylalanine (PHE) nor tyrosine (TYR) concentrations; however, the PHE/TYR ratio was significantly (*p* < 0.01) increased in the high disease activity (HDA) group compared to the clinically inactive (IA) group (see Figure 2) The lower reference values of PHE concentrations in healthy children suggest increased PHE levels in both the IA and HDA group.

### 2.3. IDO Activity

#### 2.3.1. Kynurenine and Tryptophan

Tryptophan (TRP) concentrations in the high disease activity (HDA) group were significantly (*p* < 0.01) lower than in clinically inactive (IA) patients. Consequently, the KYN/TRP ratio was significantly (*p* < 0.05) increased in the HDA group compared to the IA group. The lower reference values of KYN concentrations in healthy children suggest increased KYN levels in both the IA and HDA group (see Figure 3). 

#### 2.3.2. Kynurenine, Tryptophan, and MTX Treatment

Remarkably, MTX did not affect the KYN/TRP ratio in the IA group, but MTX significantly (*p* < 0.05) decreased the KYN/TRP ratio in the HDA group (see Figure 4). Nevertheless, the lower reference values of KYN/TRP ratio in all groups suggests increased KYN/TRP ratios in both the IA and HDA group (see Figure 4). 

## 3. Discussion

As hypothesized, the present study clearly showed that JIA patients with high disease activity had higher levels of neopterin but lower levels of tryptophan and a higher ratio of both KYN/TRP and PHE/TYR than clinically inactive JIA patients. This is in agreement with the hypothesis that inflammation increases the enzymatic activity of both IDO and GTP–CH1 and decreases the efficacy of the co-factor BH4. Interestingly, these affected enzymatic pathways, and co-factor BH4, play a critical role in the synthesis of the brain neurotransmitters noradrenaline, dopamine, serotonin, melatonin, and glutamate, which are known for their role in mental health problems, including fatigue, cognition, anxiety, and depression. Therefore, it is not surprising that JIA patients with high disease activity have decreased well-being. In Figure 5, we present the hypothetical construct of how proinflammatory cytokines affect the brain.

### 3.1. GTP–CH1 Pathway

#### 3.1.1. Neopterin

As expected, the present study shows that neopterin levels are significantly increased in JIA patients with high disease activity (cJADAS) compared to clinically inactive JIA patients. This is in line with earlier findings of Abu Shady and co-workers, showing a significant positive correlation between serum neopterin and the erythrocyte sedimentation rate (ESR), TNF-α, IL-6, monocyte chemoattractant protein-1 (MCP-1), and disease activity as measured by JADAS-27 [27]. It was shown that both serum neopterin levels and JADAS-27 reflected disease activity in JIA. In agreement, in rheumatoid arthritis (RA) patients, a significant positive correlation between neopterin levels and disease activity was observed [28]. Similar observations have been made in patients with Crohn’s disease [29] and ulcerative colitis [30]. Thus, the biomarker neopterin is not specific for arthritis. In agreement, high neopterin (6-d-trihydroxypropyl-pterin) levels have been observed during inflammation and/or autoimmunity, aging (>80 year), bacterial, viral, and parasite infections, rejection episodes after allograft transplantation, and several malignant tumor diseases [31].

There is general agreement that neopterin levels reflect the activity of the cellular immune system [31]. This can be explained as follows (see Figure 5): Activated T-helper 1 (Th1) lymphocytes produce so-called lymphokines, such as interferon-gamma (IFN-γ) [32], which in macrophages and dendritic cells stimulates the enzyme guanosine triphosphate-cyclohydrolase 1 (GTP–CH1) [33,34] (See Figure 5). Although tumor necrosis factor-α (TNF-α) does not directly stimulate GTP–CH1, it may enhance the sensitivity of macrophages to IFN-γ, and therefore further increase the activity of the GTP–CH1 pathway [35]. Consequently, 7,8-dihydroneopterin-triphosphate (NH2P3) is formed as an intermediate from GTP by GTP–CH1 [36]. Remarkably, there is almost no biopterin-forming enzyme pyruvoyl–tetrahydropterin synthase (PTPS) present in human macrophages and monocytes, in contrast to brain cells, and therefore, NH2P3 is largely used for neopterin synthesis and release by the activated immune system [36], at the expense of the synthesis of BH4 by other cells [37]. 

There is a growing body of evidence that neopterin is not only a biomarker of Th1-cellular immune activity, but neopterin also plays an important role in host–defense reactions [38]. Activation of monocytes and macrophages increases the production of reactive oxygen and nitrogen species (ROS/RNS). It has been suggested that neopterin amplifies the cytotoxic forces of ROS/RNS directed against invading pathogens, and it increases oxidative stress during immune activation [38].

#### 3.1.2. Tetrahydrobiopterin (BH4)

The higher PHE/TYR ratio in high disease activity JIA patients was in accordance with our hypothesis and supports the notion that increased inflammation in JIA patients lowered BH4 activity, which is also corroborated by higher neopterin levels. This is in agreement with earlier findings in elderly persons with chronic low-grade inflammation that inflammation was associated with increases in phenylalanine concentrations at the expense of tyrosine [23]. Furthermore, this study showed that a higher PHE/TYR ratio was correlated more with neurovegetative symptoms, including sleep disturbance, digestive symptoms, fatigue, sickness, and motor symptoms [23] (see Figure 6). 

The above described findings in JIA patients can be explained as follows: Levels of BH4 drop despite the inflammation-induced increase in GTP–CH1, which is the rate-limiting enzyme for the biosynthesis of BH4 from guanosine triphosphate (GTP), because NH2P3 is largely used for neopterin synthesis at the expense of the synthesis of BH4 [31,39]. Since JIA patients have more oxidative and nitrosative stress [40] in the presynaptic neurons, the highly redox-sensitive BH4 can easily be oxidized to the inactive dihydrobiopterin (BH2) [41] due to inflammation-induced increases in nitric oxide synthases (NOS), ROS, and RNS [17,39,42,43,44].

Importantly, BH4 is a crucial co-factor for several rate-limiting enzymes to synthesize different monoamine neurotransmitters [10]. The BH4-influenced enzymes are: 1. Phenylalanine hydroxylase (PAH) for the conversion of l-Phenylalanine (PHE) to l-Tyrosine; 2. l-Tyrosine hydroxylase (TH) for the conversion of l-Tyrosine to 3,4-Dihydroxy-l-phenylalanine (l-DOPA) and followed by conversion to dopamine and to noradrenaline; 3. Tryptophan hydroxylase (TPH) for the conversion of l-Tryptophan (TRP) to 5-hydroxytryptophan (5-HTP). 5-HTP is further converted to serotonin [11], to n-acetyl-serotonin, and to melatonin; 4. Nitric oxide synthases (NOS) produce nitric oxide (NO) by catalyzing a five-electron oxidation of a guanidino nitrogen of l-arginine [23,44,45]. Thus, in JIA patients, the oxidative and nitrosative stress (ROS/RNS) may ultimately lead to decreased BH4 availability and consequently, decreased serotonin, melatonin, dopamine, and noradrenaline levels. Especially, an inflammation-induced decrease in dopamine concentrations has been shown to be responsible for anhedonia and severe fatigue [13,23] (see Figure 6). In agreement, it has been shown that the PHE/TYR ratio was associated with fatigue and was negatively correlated with cerebrospinal fluid dopamine concentrations [8]. Furthermore, imaging studies in patients suggest that inflammation produces mental fatigue because of lower dopaminergic neurotransmission in the cortico-striatal network, including the striatum and prefrontal cortex [15,21,46].

Interestingly, the folate antagonist MTX during high disease activity (HDA) significantly reduced the KYN/TRP ratio towards more normal values compared to HDA without MTX treatment (see Figure 4), probably via the lowering of both inflammation and IDO levels (see Figure 5, at the top). Following this reasoning, it was expected that MTX treatment could improve patients’ overall mental health outcomes, because it is generally accepted that IDO levels are inversely correlated with serotonin levels. In contradiction (see Table 1), MTX treatment had a significantly worse effect on visual analogue scale (VAS) scores, when compared to HDA patients not treated with MTX. It is speculated that this apparent contradiction can be explained by the fact that MTX (like sulfasalazine) directly lowers the biosynthesis of BH4 (see Figure 5, in the midst). Consequently, MTX may attenuate the activities of the enzymes PAH, TH, and TPH, resulting in lower noradrenaline, dopamine, and serotonin concentrations, respectively. These lower monoamine concentrations may explain why MTX itself can produce depressive-like symptoms and cognitive impairment, despite the inflammation-lowering effects [47].

### 3.2. IDO Pathway

#### 3.2.1. Tryptophan, Serotonin, and Melatonin 

The lower tryptophan concentration and the higher KYN/TRP ratio in JIA patients with high disease activity suggests an increase in tryptophan breakdown and therefore supports our hypothesis that the enzymatic activity of IDO is increased in JIA patients with an increased inflammatory state compared to clinically inactive JIA patients. Our results are in agreement with a study by Capuron and coworkers showing similar changes in tryptophan and tyrosine metabolism in sera of the elderly during chronic low-grade inflammation [23]. In accordance, it has been reported that upregulation of IDO also led to a decrease in the serotonin/tryptophan ratio in the bilateral hippocampus of mice suffering from arthritis [48]. In addition, during inflammation, increased IDO activity was associated with higher microglia activity in the brain [49,50]. Remarkably, it has been reported that elevated IDO activity can be observed in patients with depression as well as in rats suffering from anhedonia [51,52,53].

It is well known that altered serotonin levels are involved in aggression and impulsivity [54,55], sleep [56], anxiety [57], and mood disorders [58]. Normally, serotonin is used for the synthesis of melatonin in the pineal gland; therefore, lowered serotonin concentrations may result in reduced melatonin levels (see Figure 6). Since melatonin plays an important role in sleep regulation, this may explain why sleep can be disturbed in JIA patients [59].

#### 3.2.2. Kynurenine, Quinolinic Acid, and Glutamate

Proinflammatory cytokines increase IDO activity and shunt tryptophan away from the serotonin route into the kynurenine route. Kynurenine, via different routes, is metabolized into either 3-hydroxykynurenine (3HKyn) and quinolinic acid (QA) in microglia or kynurenic acid (KA) in astrocytes [25,60,61]. Interestingly, Raison and co-workers found an association between IFN-α-induced synthesis of central kynurenine and QA in cerebrospinal fluid (CSF) and the development of depressive symptoms [61], whereas there was no such an association with KA [62,63]. 

Furthermore, 3HKyn generates free-radical species (including QA) that cause oxidative stress, while QA is also an *N*-Methyl-d-aspartate (NMDA) receptor agonist [64,65]. Thus, ROS/RNS may stimulate astrocytes to release glutamate and to inhibit glutamate transporters, especially excitatory amino acid transporter 2 (EAAT2) located on astrocytes [66,67]. Jointly, these actions result in increased glutamatergic neurotransmission [68] and potentially produce neurotoxic effects [69]. 

By contrast, during inflammation, KA may function as an NMDA receptor antagonist and, therefore, KA may have neuroprotective properties [70,71]. Surprisingly, endogenous KA has been shown to activate dopaminergic neurons in the rat ventral tegmental area through glutamatergic mechanisms [72]. Up to now, however, there has been no evidence that KA can compensate for the lower dopamine release due to decreased BH4 concentrations in patients. Furthermore, KA can suppress the prefrontal cortex (PFC) via presynaptic inhibition of α7-nicotinic acetylcholine receptor signaling, producing cognitive deficits (see Figure 6) [73].

### 3.3. Increased Monoamine Reuptake

Previously, it has been demonstrated that both lipopolysaccharides (LPS) and proinflammatory cytokines increase monoamine transporter (SERT, DAT, NET) trafficking and function [74,75,76,77]. There is a growing body of evidence that the influence of proinflammatory cytokines on SERT trafficking is p38 MAPK-dependent, resulting in increased SERT availability and more reuptake of serotonin [74,75,76,77,78,79,80,81,82]. Other kinases, however, may be associated with SERT regulation, such as Protein Kinase C (PKC), ERK1/2, phosphatidylinositol 3-Kinase/Akt, and adenosine [76,79,83]. In agreement, in humans, both in healthy adult women and psoriasis/psoriatic arthritis patients, a high correlation was observed between circulating inflammatory markers and brainstem SERT availability in humans. As expected, TNF-α inhibition with etanercept reduced this SERT availability [84]. More about immune activity and monoamine transporter function can be found in the present Special Issue of *Pharmaceuticals* [85].

### 3.4. Pharma-Food Interventions

#### 3.4.1. Inflammation and Immunosuppressants

The first and most logic approach is the use of immunosuppressant drugs, since inflammation is the first link in the neuroimmune-cascade resulting in fatigue, cognitive impairment, and depressive symptoms (e.g., anhedonia), In agreement, TNF-α antibodies (e.g., infliximab) improve sleep quality [86] and depressive symptoms in patients with high baseline inflammatory biomarkers [87].

#### 3.4.2. Monoamines, Transporters, and Monoamine Reuptake Inhibitors

Since inflammation increases the biological activity of monoamine transporters, the second approach is the use of specific serotonin reuptake inhibitors (SSRIs), serotonin norepinephrine reuptake inhibitors (SNRIs), and dopamine norepinephrine reuptake inhibitors (DNRIs) in order to restore the shortage of monoamines in the synaptic cleft [58].

#### 3.4.3. Monoamines, BH4, and Nutritional Interventions

Regarding nutritional interventions, one has to be very critical because the European Food Safety Authority (EFSA) only awarded a few health claims of nutrition. Theoretically, one might expect positive effects of antioxidants (vitamin C and vitamin E), because these supplements may reduce oxidative stress and therefore counteract the oxidation of BH4. In agreement, it has been suggested to use antioxidants to restore BH4 function in the treatment of inflammation-related cardiovascular disease [88,89].

Since the inflammation-induced reduction in BH4 levels may ultimately result in lower melatonin concentrations, the use of melatonin may counteract this deficit. In addition, melatonin is a powerful antioxidant. Therefore, it has been speculated that melatonin might protect BH4 from oxidation [90]. More research, however, is needed to investigate the role of melatonin in arthritis.

There are only analogues of BH4 commercially available that are very expensive. Therefore, they cannot be used as a general supplement. BH4 concentrations can alternatively be increased by nutritional interventions, such as the use of folate (B_9_-vitamin) and the biologically active metabolite 5-methyltetrahydrofolate (5-MTHF, l-methylfolate). They are involved in the re-methylation of homocysteine, creating methionine. The downstream metabolite of methionine is S-adenosylmethionine (SAMe), which is also a methyl donor that stimulates the synthesis of BH4. Therefore, folate, l-methylfolate, and SAMe can be used to counteract the inflammation-induced decrease in BH4. In agreement, there is a growing body of evidence that l-methylfolate and SAMe supplementation might improve antidepressant actions of SSRIs in patients, who normally tend not to respond well to SSRI drugs [65,91,92]. 

#### 3.4.4. Glutamate and Nutritional Interventions

Inflammation and chronic stress increase glutamatergic neurotransmission [65,93] but decrease expression of brain-derived neurotrophic factor (BDNF) in limbic structures [94] and consequently alter dendritic remodeling in subareas of PFC, the amygdala, and the hippocampus, decrease the number of hippocampal dendritic spines and decrease dentate gyrus neurogenesis (for review: See Reference [95]). Interactions between the corticosteroid (their receptors) and glutamate systems play a pivotal role in the altered neuroplasticity and development of mental disorders [95]. Recently, in a chronic stress animal model with sustained activation of neural mineralocorticoid receptors, it was reported that acetyl-l-carnitine (LAC), via an epigenetic mechanism (i.e., acetylation), induced mGlu2 receptors in the hippocampus and PFC [96,97]. Furthermore, LAC treatment normalized both glutamate overflow and increased BDNF expression [96,97,98]. Decreased plasma LAC levels have been observed in patients suffering from major depression, especially in those with stronger severity, earlier disease onset, and treatment-resistant depression [99]. In agreement, in animal models of depression, LAC leads to antidepressant-like responses after a few days of administration [96,97,98]. Therefore, it is speculated that LAC treatment, via normalizing glutamate overflow and BDNF expression, may have fast positive effects on the neuropsychiatric symptoms in juvenile arthritis patients.

Increased glutamatergic neurotransmission can also be normalized by a ketogenic diet. A ketogenic diet is a low-carb diet often with ketone bodies (acetoacetate and β-hydroxybuturate) or medium-length fatty acids generating ketone bodies. Ketone bodies are thought to inhibit mitochondrial production of ROS following glutamate excitotoxicity by increasing NADH oxidation [100]. A ketogenic diet has profound effects on multiple targets, including but not limited to, lowering glutamate- and increasing GABA transmission, and lowering oxidative stress [101]. Therefore, it is speculated that a ketogenic diet may also have positive effects on the supposed imbalance between monoamines and glutamate in the emotional brains of juvenile arthritis patients. 

## 4. Materials and Methods 

### 4.1. Study Population

Twenty-four serum samples of anonymous children, between 8 and 18 years of age, which were diagnosed with JIA were collected. All participants were diagnosed with JIA. Disease activity was displayed by the clinical juvenile arthritis disease activity score (cJADAS). The cJADAS is constructed around three elements: The active joint count (AJC), physician global assessment (PGA), and parent/patient visual analogue scale (VAS) of well-being [102]. In more detail, the parent/patient is asked to make a global assessment of the child’s overall well-being on a 10 cm visual analogue scale, with anchors of ‘0 = very good’ and ‘10 = very poor’. The cJADAS score has shown to be a very useful tool to identify JIA patients with high disease activity in need of anti-TNF therapy [103]. 

The Medical Ethics Committee (METC) of the Utrecht University Medical Centre (UUMC), the Netherlands, approved the present study, as part of project PharmaChild METC number 11-499/C.

Fifteen participants were female (62.5%) and nine participants (37.5%) were male. Patients were grouped based on disease activity (high disease activity (i.e., cJADAS ≥10.0) or clinically inactive (i.e., cJADAS <0.5). The corresponding CRP values were in line with the disease activity scores. Half of the patients were treated with MTX; the other half of the patients did not use any immunosuppressants. Since there was no difference in disease activity between male and female participants, the JIA-patients were divided in four groups (*n* = 6 per group) based on disease activity and MTX use (see Table 1).

### 4.2. Blood Samples

Blood samples for the measurement of tryptophan, tyrosine phenylalanine, kynurenine, and neopterin were stored at −80 °C at Wilhelmina Children’s Hospital of University Medical Center Utrecht and two weeks before conductance of the assays moved to a lab freezer at −20 °C. Serum concentrations of neopterin and phenylalanine were measured by enzyme-linked immunosorbent assay (Tecan, Giessen, the Netherlands; and Immundiagnostik AG, Bensheim, Germany). Free tryptophan, tyrosine, and kynurenine serum concentrations were determined by high-performance liquid chromatography with electrochemical detection (Antec, vt-03 flow cell with ISAAC reference electrode). The potential of the detector was set at 0.9 V. The column was a NeuroSep C18 (Antec, NeuroSep 115 C18, 1 mm × 150 mm, 3 μm particle size). The mobile phase consisted of 10% Methanol, 50 mM phosphate buffer pH 2.5, 0.1 mM EDTA, and 8 mM KCl and was pumped at 50 µL/min through the column. The autosampler temperature was set at 8 °C degrees. To serum samples, methyldopa was added as internal standard. The samples were deproteinated using perchloric acid, centrifuged, and the supernatant was mixed with potassium acetate (equimolar to perchloric acid) to reduce acidity. The samples were centrifuged once more to remove the precipitate and the supernatant was diluted eight times with ultra-pure water. A total of 5 µL was subsequently injected. Calibration curve samples and quality control samples were prepared in bovine serum albumin solution and pretreated and analyzed as described for serum samples. Precision (rsd) and accuracy (deviation from actual concentration) of the method were below 10% for all analytes and calibration curve concentrations. The ratios of KYN/TRP and Phe/TYR were used as indexes of, respectively, IDO activity and GTP–CH1 activity.

### 4.3. Statistics

All neopterin and phenylalanine concentrations were measured in threefold. In two participants, one of the three neopterin measurements was not considered for data analyses, because these were completely out of range and therefore treated as outliers. No missing values for tryptophan, tyrosine, and kynurenine were obtained. Concentrations of biological markers were compared between the two groups, i.e., clinically inactive patients (IA) and high disease activity (HDA), with the independent samples *t*-test. Last, Bonferroni’s post hoc analysis was used to compare the biological markers across the four subgroups (see Table 1: IA; HDA; without MTX; and with MTX). As a reference group, concentrations of biomarkers of healthy children from previous studies were used [24,25,26]. These were only used as a rough indication and were not used for statistics, because both age and gender distribution differed from our JIA patients. All statistical analyses were performed with SPSS21. Data are presented as means ± SEM. The significance level for all tests was set a *p* < 0.05.

## 5. Conclusions

JIA patients with high disease activity are characterized by higher serum levels of both neopterin and kynurenine, and a higher ratio of both KYN/TRP and PHE/TYR and lower tryptophan serum levels than in clinically inactive JIA patients, reflecting increased activity of both IDO and GTP–CH1 pathways but decreased BH4 efficacy during chronic inflammation. Here, it is hypothesized that this may produce an imbalance between glutamate and monoamines with negative implications for well-being, including fatigue, cognition, anxiety, and depression.

## Figures and Tables

**Figure 1 pharmaceuticals-12-00009-f001:**
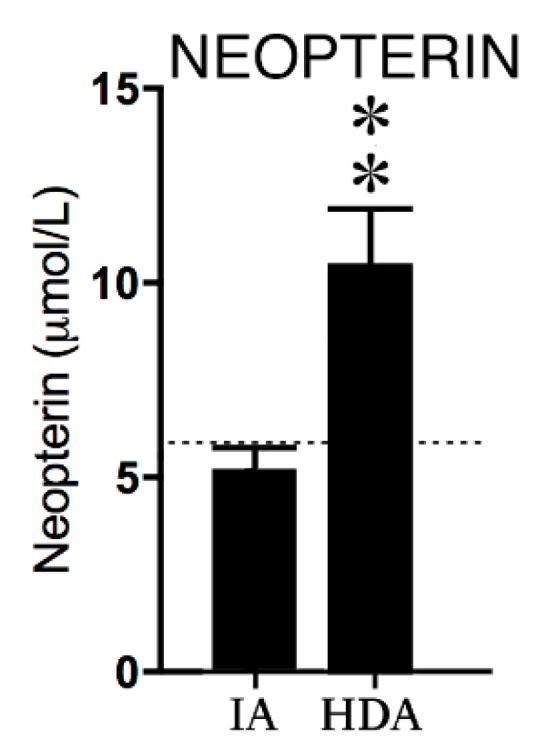
Neopterin concentrations in clinically inactive patients (IA) (*n* = 12) and high disease activity (HDA) (*n* = 12). The dotted line is the reference value in healthy children from previous studies [24,25,26]. Data are presented as means ± SEM. Significance is described as ** *p* < 0.01.

**Figure 2 pharmaceuticals-12-00009-f002:**
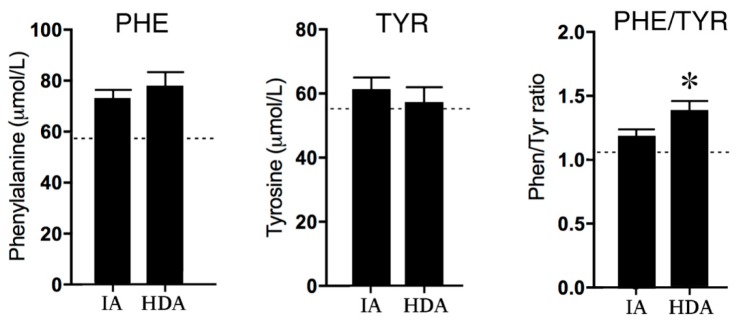
Phenylalanine (PHE) and Tyrosine (TYR) concentrations and the PHE/TYR ratio in clinically inactive patients (IA) (*n* = 12) and high disease activity (HDA) (*n* = 12). The dotted lines are the reference values in healthy children from previous studies [24,25,26]. Data are presented as means ± SEM. Significance is described as * *p* < 0.05.

**Figure 3 pharmaceuticals-12-00009-f003:**
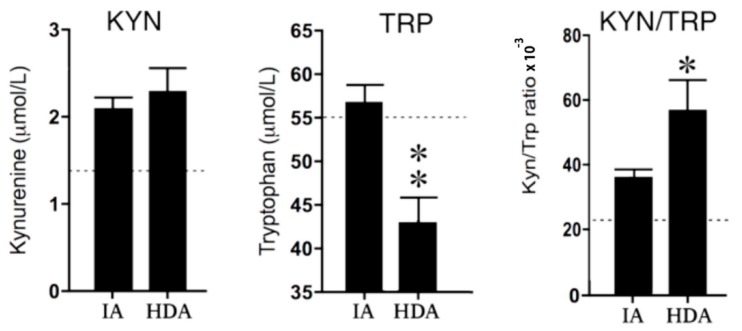
Kynurenine (KYN) and tryptophan (TRP) concentrations and the KYN/TRP ratio × 10^−3^ in clinically inactive patients (IA) (*n* = 12) and high disease activity (HDA) (*n* = 12). The dotted lines are the reference values in healthy children from previous studies used [24,25,26]. Data are presented as means ± SEM. Significance is described as * *p* < 0.05 and ** *p* < 0.01.

**Figure 4 pharmaceuticals-12-00009-f004:**
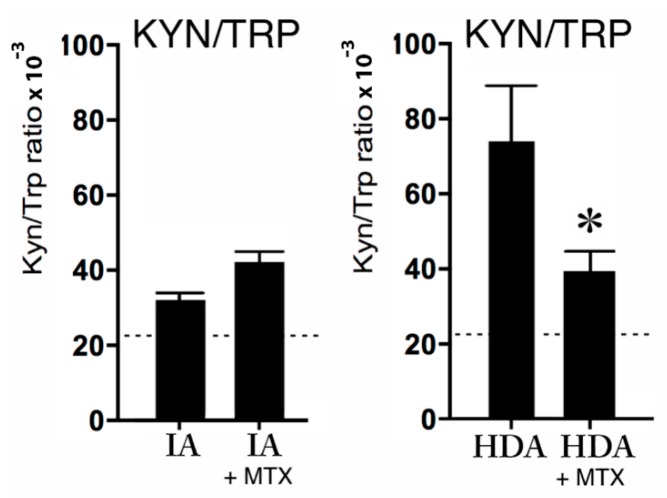
The KYN/TRP ratio × 10^−3^ in clinically inactive patients (IA) without (*n* = 6) or with MTX treatment (*n* = 6), and the KYN/TRP ratio × 10^−3^ during high disease activity (HDA) (*n* = 6) without or with MTX treatment (*n* = 6). The dotted lines are the reference values in healthy children from previous studies [24,25,26]. Data are presented as means ± SEM. Significance is described as * *p* < 0.05.

**Figure 5 pharmaceuticals-12-00009-f005:**
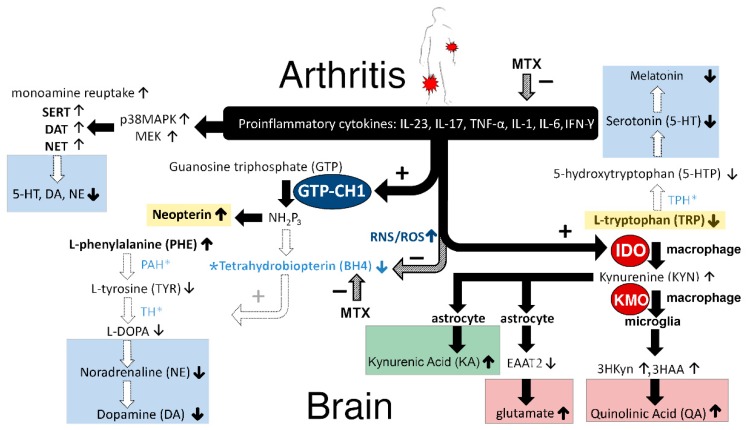
Hypothetical construct of how proinflammatory cytokines affect neurotransmitters in the brain (explanation of the figure can be found in the Discussion section).

**Figure 6 pharmaceuticals-12-00009-f006:**
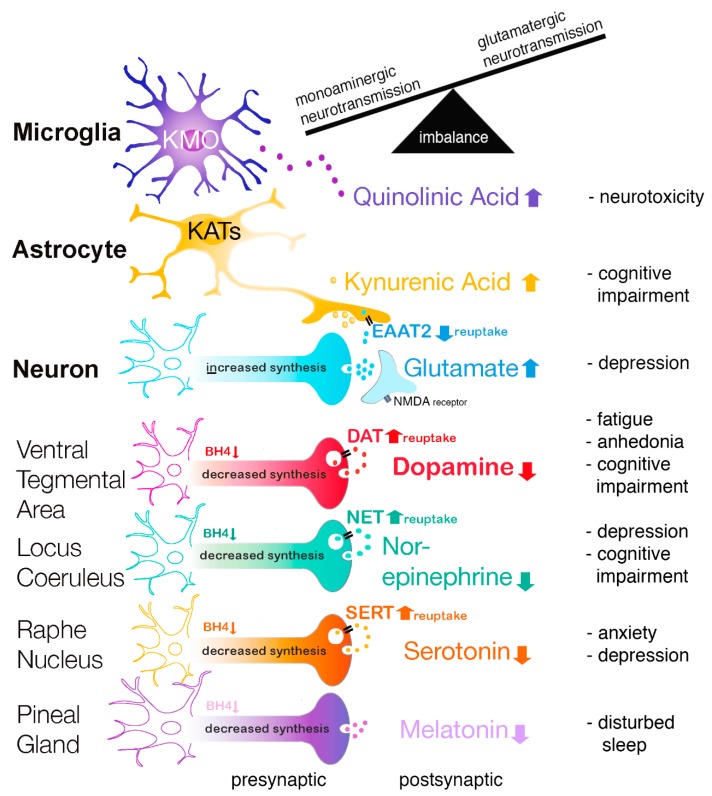
Relation between inflammation, neurotransmitter imbalance, and brain disorders.

**Table 1 pharmaceuticals-12-00009-t001:** The juvenile idiopathic arthritis (JIA) patients were divided into four groups based on disease activity and methotrexate (MTX) use. Data are presented as means ± SEM. Significance is described as * *p* < 0.05 for comparison of high disease activity versus clinically inactive and ^#^
*p* < 0.05 for comparison of high disease activity without MTX versus high disease activity with MTX.

*DISEASE ACTIVITY*	High Disease Activity (HDA)	High Disease Activity (HDA)	Clinically Inactive (IA)	Clinically Inactive (IA)
***MTX***	No	Yes	No	Yes
*cJADAS*	18.7 ± 2.5 *	14.3 ± 0.4 *	0	0
*CRP*	90.2 ± 38.8 *^#^	31.8 ± 10.6 *	0.3 ± 0.1	0.1 ± 0.1
*VAS*	4.98 ± 1.26 *	6,82 ± 0.9 *	0.0 ± 0.0	0.08 ± 0.04

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
