# Peer review of "Juvenile Arthritis Patients Suffering from Chronic Inflammation Have Increased Activity of Both IDO and GTP-CH1 Pathways But Decreased BH4 Efficacy: Implications for Well-Being, Including Fatigue, Cognitive Impairment, Anxiety, and Depression"

_pharmaceuticals, 2019, doi:10.3390/ph12010009_

Reviewer 1 Report

The authors have reported on a well-designed study to assess levels of biochemical markers of mental well-being in patients with juvenile idiopathic arthritis. The key findings were that enzymatic markers of IDO and GTP-CH1 were elevated in patients who displayed high disease activity and the authors have suggested altered brain neurotransmitters as a result (supplemented with VAS data).

Overall, the study is well written and presented and is of excellent scientific standard. In spite of this, I have a few comments/recommendations.

With regards to the representation of ratios in graphs (example figure 3, KYn/Trp) the values of the Y axis are uninformative. I highly recommend the authors to express the mean values of Kyn and TRP as fold changes relative to controls and from this determine the ratio (where the KYN/TRP value of 1 represents an equal portion of KYN and TRP relative to the controls).

Figure 4, specifically the KYN/TRP ratio in patients with HDA (where there without MTX) is particularly interesting however was not addressed in the discussion. Although the lower reference value of KYN/TRP ratio suggest an overall increase in IDO activity in patients with IA an HDA, one cannot discount the significant decrease in the KYN/TRP ratio in patients with HDA receiving MTX, when compared to HDA patients with no MTX treatment.

This would suggest that MTX significantly attenuates IDO levels in patients with HDA and following the authors logic (and the dogma within the literature), MTX treatment could improve patient's overall mental health outcomes. This is based on the many evidences that IDO levels are inversely correlated with serotonin levels. Despite this, the authors have reported that HDF patients with MTX treatment have a significantly worse VAS scores when compared to HDF patients not treated with MTX. This alone suggests that there is no dose-dependent relationship of IDO in relation to changes in mental well-being.

I would strongly suggest that the authors include a paragraph in their discussion that addresses figure 4 and the above argument.

Author Response

We thank the reviewer for her/his very constructive comments, and we fully agree with all the points that have been raised. Accordingly, we have revised the manuscript.

The English language and style were checked. The changes are shown in blue in the revised manuscript.

 With regards to the representation of ratios in graphs (example figure 3, KYn/Trp) the values of the Y axis are uninformative. I highly recommend the authors to express the mean values of Kyn and TRP as fold changes relative to controls and from this determine the ratio (where the KYN/TRP value of 1 represents an equal portion of KYN and TRP relative to the controls).

Response:

Since we do not have the individual data of controls from litterature, we can not do this. After careful checking all the data we discovered that the ratio had to be multiplied by x10-3, because we made a mistake (Kyn was presented in nmol/L, whereas it should have been µmol/L). Now the data are fully in line with all the values of Kyn/Trp found in litterature

Figure 4, specifically the KYN/TRP ratio in patients with HDA (where there without MTX) is particularly interesting however was not addressed in the discussion. Although the lower reference value of KYN/TRP ratio suggest an overall increase in IDO activity in patients with IA an HDA, one cannot discount the significant decrease in the KYN/TRP ratio in patients with HDA receiving MTX, when compared to HDA patients with no MTX treatment.

This would suggest that MTX significantly attenuates IDO levels in patients with HDA and following the authors logic (and the dogma within the literature), MTX treatment could improve patient's overall mental health outcomes. This is based on the many evidences that IDO levels are inversely correlated with serotonin levels. Despite this, the authors have reported that HDF patients with MTX treatment have a significantly worse VAS scores when compared to HDF patients not treated with MTX. This alone suggests that there is no dose-dependent relationship of IDO in relation to changes in mental well-being. 

I would strongly suggest that the authors include a paragraph in their discussion that addresses figure 4 and the above argument.

 Response:

We fully agree with the reviewer that the above argument had to be addressed, therefore a new paragraph was included in the discussion.

added: line 264-276;

Interestingly, the folate antagonist MTX during high disease activity (HDA) significantly reduced KYN/TRP ratio towards more normal values, as compared to HDA without MTX treatment (see Figure 4.), probably via the lowering of both inflammation and IDO levels (see Figure 5., at the top). Following this reasoning, it was expected that MTX treatment could improve patient's overall mental health outcomes, because it is generally accepted that IDO levels are inversely correlated with serotonin levels. In contradiction (see Table 1.), MTX treatment had a significantly worse effect on VAS scores, when compared to HDA patients not treated with MTX. It is speculated, that this apparent contradiction can be explained by the fact that MTX (like sulfasalazine) directly lowers the biosynthesis of BH4 (see Figure 5., in the midst). Consequently, MTX may attenuate the activities of the enzymes PAH, TH, and TPH, resulting in lower noradrenaline, dopamine and serotonin concentrations, respectively. These lower monoamine concentrations may explain why MTX itself can produce depressive-like symptoms and cognitive impairment, despite the inflammation lowering effects [47].

Reviewer 2 Report

In this manuscript authors investigate the outcome of inflammation in GTP-CH1 and IDO-pathway in arthritis  (8-18yrs old) cases. The reported results are interesting. I have following comments:

a. In all the figure legends, authors have not reported whether the data were Mean+-SEM/SD? This info is lacking in Statistical Analysis part as well. Authors should address this issue.

b. Authors are required to show the Th1/Th17-related cytokines levels expression on the serum samples from the test and control cases. This issue has been addressed in animal models with IDO Knock out mice in published literatures, but it would be interesting to see the role of IDO on modulation of these cytokines in humans.

c. In fig. 2, the Phen/Tyr ratio between IA and HDA group does not seem to be promising? Is it statistically significant considering Mean+-SEM/SD? Also authors need to show which statistical test they utilized to generate the significance in the bar diagram. It appears that there is a biological significane rather than statistical. Authors need to double check on this.

d. Fig. 3, the middle bar diagram has two asterisks but the figure legend lacks why authors put those multiple stacking symbols on the fig.

e. Fig. 5, the fig seems to be complex. Can it be broken down into two?

Author Response

We thank the reviewer for her/his very constructive comments, and we fully agree with all the points that have been raised. Accordingly, we have revised the manuscript.

The English language and style were checked. Furthermore, methods description was improved, and the results were presented more clearly. The changes are shown in blue in the revised manuscript.

a. In all the figure legends, authors have not reported whether the data were Mean+-SEM/SD? This info is lacking in Statistical Analysis part as well. Authors should address this issue.

Response: This has been made clear in the revised manuscript: line 130, line 142, lines 158-159, line 171, and also in Statistical Analysis part, line 455 : “Data are presented as means ± SEM”.

b. Authors are required to show the Th1/Th17-related cytokines levels expression on the serum samples from the test and control cases. This issue has been addressed in animal models with IDO Knock out mice in published literatures, but it would be interesting to see the role of IDO on modulation of these cytokines in humans.

Response: We fully agree that it is interesting to study Th1/Th17, bus also T regulatory cells. But this will be done in future studies. Therefore, in the present manuscript, we mostly talk about inflammation or proinflammatory cytokines in general. The high CRP and cJADAS values confirm the presence of inflammation. In the past, many studies have shown that JIA patients, with high disease activity, have increased concentrations of proinflammatory cytokines. But indeed, in future studies it would be very interesting whether there are correlations between some of the specific proinflammatory cytokines and biomarkers, as measured in the present manuscript.

c. In fig. 2, the Phen/Tyr ratio between IA and HDA group does not seem to be promising? Is it statistically significant considering Mean+-SEM/SD? Also authors need to show which statistical test they utilized to generate the significance in the bar diagram. It appears that there is a biological significane rather than statistical. Authors need to double check on this.

Response:  We fully agree with the reviewer that these sections have to be made clearer and more discussed. Therefore, in the revised manuscript description of statistical test has been made clearer: line 448-451; “Concentrations of biological markers were compared between the two groups, i.e., clinically inactive patients (IA) and high disease activity (HDA), with the independent samples t-test. Last, Bonferroni’s post hoc analysis was used to compare the biological markers across the four subgroups (see table 1: IA; HDA; without MTX; and, with MTX).”

The biological significance was discussed: line 227-233;” The higher PHE/TYR ratio in high disease activity JIA patients was in accordance with our hypothesis and supports the notion that increased inflammation in JIA patients lowered BH4 activity, which is also corroborated by higher neopterin levels. This is in agreement with earlier findings in elderly persons with chronic low-grade inflammation that inflammation was associated with increases in phenylalanine concentrations at the expense of tyrosine [23]. Furthermore, this study showed that higher PHE/TYR ratio was correlated more with neurovegetative symptoms, including sleep disturbance, digestive symptoms, fatigue, sickness, and motor symptoms [23] (see Figure 6.).”

d. Fig. 3, the middle bar diagram has two asterisks, but the figure legend lacks why authors put those multiple stacking symbols on the fig.                                                                                                           

Response: This has been made clear in the revised manuscript: line 158-159; “Data are presented as means ± SEM. Significance is described as * p < 0.05 and ** p < 0.01.”

e. Fig. 5, the fig seems to be complex. Can it be broken down into two?

Response:  We agree that the figure is complex, but neuroimmune interactions are complex. By showing the entire overview at once, the different connections are more transparent.